# Molecular Characterization of Anion Exchanger 2 in *Litopenaeus vannamei* and Its Role in Nitrite Stress

**DOI:** 10.3390/ijms26030964

**Published:** 2025-01-23

**Authors:** Xuenan Li, Xilin Dai

**Affiliations:** 1Key Laboratory of Freshwater Aquatic Genetic Resources, Ministry of Agriculture and Rural Affairs, Shanghai Ocean University, Shanghai 201306, China; a15165221580@163.com; 2Shanghai Collaborative Innovation for Aquatic Animal Genetics and Breeding, Shanghai 201306, China; 3National Experimental Teaching Demonstration Centre for Aquatic Sciences, Shanghai Ocean University, Shanghai 201306, China

**Keywords:** *Litopenaeus vannamei*, *AE2*, nitrite stress, tissue localization, RNA interference

## Abstract

Anion exchanger 2 (AE2) mediates the Cl^−^/HCO_3_^−^ transmembrane exchange process and regulates intracellular pH homeostasis. In this study, the *AE2* gene (GenBank: PQ073349) was cloned and characterized from *Litopenaeus vannamei* using the rapid amplification of cDNA ends (RACE) technique. Employing bioinformatics, real-time fluorescence quantitative PCR, and RNA interference, we explored the gene’s sequence characteristics, tissue distribution, and the effects of nitrite on shrimp survival, physiology, and tissue damage following gene silencing. The results showed that AE2 cDNA was 5134 bp in length, encoding 1293 amino acids, which includes both the Band3 and HCO_3_^−^ structural domains. *AE2* was expressed in all tissues, with the highest expression in muscle. After silencing *AE2*, shrimp survival increased and hemolymph nitrite levels decreased. Notably, the oxidative stress enzyme system was not severely affected, and gill tissue damage was reduced. In addition, the expression level of Na^+^/K^+^/2Cl^−^ cotransporter 1 (*NKCC1*) was significantly reduced (*p* < 0.05). These findings suggest that AE2 and NKCC1 are jointly involved in regulating the physiological process of nitrite entry into the shrimp body through gill tissue. Overall, this study provides a crucial experimental foundation for addressing the toxicity concerns associated with nitrite.

## 1. Introduction

*Litopenaeus vannamei* exhibits advantages such as a rapid growth rate, a short culture cycle, and a broad range of salt tolerance, making it one of the major economic shrimp species worldwide [1]. Nitrite is a prominent stress factor in shrimp aquaculture environments [2]. The accumulation of residual feed, feces, and inadequate water circulation during shrimp farming can result in elevated nitrite concentrations, negatively impacting shrimp survival [3]. Increased levels of nitrite in the environment decrease the oxygen affinity of the hemolymph, thereby disrupting the normal oxygen-carrying capacity and ultimately leading to shrimp asphyxiation [4,5]. In addition, elevated nitrite concentrations undermine cell membrane stability and enzyme activity while posing a threat to the osmotic balance of the excretory system [6,7]. Therefore, in shrimp aquaculture, especially in intensive freshwater aquaculture modes, nitrite has become one of the main factors restricting the growth and survival of shrimp [8,9,10,11]. Salinity is a critical variable that influences nitrite toxicity [12], with higher salinity levels enhancing shrimp tolerance to nitrite [13]. It is widely acknowledged that salinity influences the toxicity of nitrite because nitrite ions (NO_2_^−^) and chloride ions (Cl^−^) in the water column compete for uptake sites in gill [14].

Anion exchangers (AEs) are crucial for the maintenance of intra- and extracellular ion homeostasis, cellular volume regulation, and the intracellular concentrations of Cl^−^ and bicarbonate (HCO_3_^−^), as well as pH balance in vivo. This is achieved through the reversible transmembrane exchange of Cl^−^/HCO_3_^−^ [15]. The AE family consists of three members: AE1, AE2, and AE3. Under physiological conditions, AE1 is primarily expressed in erythrocytes and mediates the transmembrane exchange of Cl^−^/HCO_3_^−^ in these cells. AE2 is widely expressed across various tissues and serves as a crucial regulator of intracellular pH homeostasis. AE3 is predominantly located in the brain and heart, where it plays a key role in regulating intracellular pH in excitable tissues [16,17]. The mechanism of ion uptake in freshwater fish and crustaceans is thought to involve the H^+^-ATPase located in the apical membrane of epithelial cells. This enzyme extrudes H^+^ ions, facilitating the entry of Na^+^ into the cell through sodium channels. The excretion of H^+^ primarily occurs through the hydration of carbon dioxide, a process catalyzed by carbonic anhydrase within the epithelial cells. The HCO_3_^−^ produced serves as a counterbalancing ion for Cl^−^ uptake via the apical Cl^−^/HCO_3_^−^ exchange mechanism [15,18] (Figure 1). NO_2_^−^ exhibits a significant affinity for the chloride uptake mechanism in the gill, resulting in its accumulation in plasma as it competes with chloride and occupies chloride uptake sites. This competitive process facilitates the translocation of nitrite across the gill barrier. It is generally accepted that NO_2_^−^ replaces Cl^−^ in the gill of freshwater fish, allowing for its entry into the body via the Cl^−^/HCO_3_^−^ exchanger (AE) [19].

The mechanisms of competition between NO_2_^−^ and Cl^−^ in the gill have primarily been investigated in freshwater fish and remain poorly understood in crustaceans. In this study, the full-length cDNA sequence of *AE2* gene was cloned and obtained for the first time, using *L. vannamei* as the research object. Preliminary investigations revealed that the *AE2* gene is involved in the regulatory processes of nitrite uptake, indicating its crucial role in the entry of NO_2_^−^ into the shrimp’s body. These findings of this study will provide essential data to support the argument for the reduction in nitrite toxicity.

## 2. Results

### 2.1. AE2 Gene Sequence Analysis

The cDNA sequence of the *AE2* gene is 5134 bp in length, with an open reading frame (ORF) length of 3882 bp that encodes 1293 amino acids. The lengths of the 5′ untranslated regions (5′-UTR) and 3′ untranslated regions (3′-UTR) are 584 bp and 668 bp, respectively (Figure 2A). The gene accession number is PQ073349. The protein has a molecular weight of 144.63 kDa, an isoelectric point of 6.96, no signal peptide at the N-terminus, and contains twelve transmembrane regions (Figure 2B). Conserved structural domain prediction revealed that the gene contains Band3 and HCO_3_^−^ structural domains (Figure 2C).

The results of the homology comparison showed that the amino acid homology between AE2 in *L. vannamei* and homologs in other crustaceans was high. The highest homology was observed with *Penaeus indicus*, reaching 95.91%, followed by *Penaeus chinensis,* at 95.88% (Figure 2D).

A phylogenetic tree was constructed using the software MEGA v7.0.26 based on a homology analysis of the amino acid sequence of the *AE2* gene. The results showed that the phylogenetic tree is mainly divided into two clades: the first for shrimp and the second for crabs. *L. vannamei* belongs to the first branch and is clustered with *P. indicus*, followed by *P. chinensis.* However, *L. vannamei* is distant from *Macrobrachium nipponense* and *Halocaridina rubra* (Figure 2E).

### 2.2. Tissue Expression of AE2 Gene and Expression Analysis After Nitrite Stress

The results of real-time quantitative PCR (RT-qPCR) showed that the *AE2* gene was expressed in all tissues, with a significantly higher expression in the muscle than in other tissues (*p* < 0.05). The second highest expression was observed in the eyestalk and stomach, while the lowest expression was observed in the heart (Figure 3A).

Compared with the control group, the expression of the *AE2* gene was upregulated after 48 h of nitrite stress (*p* < 0.05). After 72 h of stress, the expression level of *AE2* reached its peak, 1.87 times higher than in the control group. Subsequently, expression was downregulated and returned to its normal level at 96 h (Figure 3B).

### 2.3. In Situ Hybridization

In this study, in situ hybridization was used to detect the distribution of AE2 mRNA in the gill of *L. vannamei*. The results showed that in the experimental group, there were obvious blue-purple hybridization signals in the gill tissue, mainly located in the cuticle and gill epithelial cells. There was no signal in the control group (Figure 4).

### 2.4. Nitrite Stress Experiments After Small Interfering (siRNA) Interference

The interference efficiency of the G1 injection was 53.1%, 54.4%, 82.2%, and 90.6% at 24, 48, 72, and 96 h, respectively. After the G2 injection, the interference efficiencies were 65.0%, 81.1%, 86.1%, and 96.5% at each time point, respectively. After the injection of G3, the interference efficiencies at each time point were 66.0%, 51.1%, 85.1%, and 84.7%, respectively (Figure 5A). After comparison, the G2 chain exhibited the best interference effect, and the G2 was selected for subsequent experiments.

After G2 injection, we re-examined the expression level of the *AE2* gene after 24 h. The results showed that the interference efficiency reached 48.5% compared to the diethyl pyrocarbonate (DEPC) group (Figure 5B). After 24 h of silencing, the shrimp were subjected to nitrite stress, and mortality rates were recorded at 24, 48, 72, and 96 h of stress. At 48–96 h, the survival rate of the interference group was significantly higher than that of the control groups. At 96 h, the survival rate was 71.7% in the interference group and 51.7% in the DEPC group (Figure 5C). The results of nitrite content in hemolymph indicated a significant reduction after interference with *AE2* (*p* < 0.05, Figure 5D).

The effects of post-silencing nitrite stress on acid phosphatase (ACP), alkaline phosphatase (AKP), superoxide dismutase (SOD), and glutathione S-transferase (GST) activities in gill tissue were examined. The results indicated that ACP activity in the control groups initially increased, followed by a subsequent decline. In contrast, ACP activity in the interference group exhibited a gradual decline. AKP activity gradually decreased in the control groups and increased in the interference group. In the control groups, GST and SOD activities first increased and then decreased, and the enzyme activities reached the maximum at 72 h. In the interference group, GST activity gradually increased, while SOD activity initially increased, then decreased, ultimately returning to normal levels (Figure 6A). After *AE2* was silenced, the expression level of the Na^+^/K^+^/2Cl^−^ cotransporter 1 (*NKCC1*) gene was gradually reduced and reached its lowest at 72 h. The expression levels of the *ACP*, *AKP,* and *SOD* genes were basically consistent with the trends in enzyme activities (Figure 6B).

### 2.5. Microscopic Observation of Gill Organization

After 24 h of nitrite stress, both the NC and interference groups began to show a swelling of hemocytes and a detachment of cuticle and epithelial cells. After 96 h of stress, gill filaments in the NC group were shriveled and deformed, more eosinophilic substances were secreted, and the cuticle structure was broken. In the interference group, the gill filaments were still neatly arranged, and there was no broken cuticle or secretion of eosinophilic substances (Figure 7).

## 3. Discussion

In this study, we first identified the *AE2* gene from *L. vannamei* and determined the accuracy of its nomenclature through phylogenetic analysis. AE2 has two typical structural domains, including the Band3 structural domain and the HCO_3_^−^ structural domain. The Band3 structural domain has the capacity to bind to a number of different proteins, including anchor proteins, hemoglobin, cytoskeletal proteins, and glycolytic enzymes. It performs multiple functions, such as stabilizing the cytoskeleton and regulating and protecting glycolytic enzymes. The HCO_3_^−^ domain is responsible for anion exchange and contains the binding site for carbonic anhydrase II, which hydrates with CO_2_ to form HCO_3_^−^ and H^+^ to regulate pH [20,21,22]. The results of the multiple sequence comparison showed that the amino acid sequence encoded by the *AE2* gene exhibited the highest degree of homology with *P. indicus* and *P. chinensis* sequences, reaching 95.91% and 95.88%, respectively. Phylogenetic tree analysis showed that *L. vannamei* was clustered with *P. indicus* and *P. chinensis* sequences. These results collectively indicate that the gene in question exhibits a high degree of conservation across crustacean species.

AE2 is the most widely distributed anion exchanger, expressed in various epithelial cells, including gastric wall cells, colon surface enterocytes, and renal collecting ducts [23]. The higher expression of *AE2* in stomach tissue is consistent with the description of the above results. In addition, the highest level of expression was observed in muscle tissue. There is evidence that Cl^−^ in muscle cells regulates intracellular pH and cell volume, as well as influencing membrane potential and contractility [24]. The primary mechanism for achieving elevated Cl^−^ concentrations is through Cl^−^/HCO_3_^−^ anion exchange [25]. Therefore, we hypothesize that AE2 plays an important role in the physiological processes of shrimp muscle. The expression level of *AE2* was low in gill tissue. However, after nitrite stress, the expression level of *AE2* increased rapidly. This is consistent with the trend in Cl^−^/HCO_3_^−^ exchanger expression after saline–alkaline stress in *Gymnocypris przewalskiii* [26]. We speculate that *AE2* does not present a high expression level under normal aquatic conditions, whereas its elevated expression under environmental stress conditions plays a critical regulatory role. In situ hybridization results indicated that positive signals were mainly concentrated in the gill epithelium and cuticle. The gill epithelium and cuticle serve as important barriers for external inorganic ions to enter the hemolymph and are also the primary regulators of osmotic pressure and hemolymph ion concentration in crustaceans [27,28].

To further validate the function of the *AE2* gene in *L. vannamei*, we employed RNAi to silence the *AE2* gene. The survival rate of the experimental group was significantly higher than that of the control group following RNAi of *AE2*. Additionally, the nitrite content in the hemolymph was significantly reduced. It has been demonstrated in related studies that nitrite uptake in freshwater scleractinian fishes primarily occurs through the active transport mechanism of chloride in the gill, which involves effective competition between NO_2_^−^ and Cl^−^. Freshwater fish and crustaceans exhibit a high level of osmotic activity compared to their surrounding environment, necessitating active ion uptake through the gill to compensate for the loss in urine and the passive outflow of ions through the gill [29,30]. Therefore, siRNA interference experiments can tentatively indicate that AE2 plays an important regulatory role in NO_2_^−^ uptake in shrimp. In addition, we observed that similar results occurred after silencing of *AE2* as were seen after silencing of *NKCC1* [31]. Silencing both genes effectively reduced hemolymph nitrite levels and reducing shrimp mortality. Therefore, we examined the changes in the expression level of *NKCC1* gene following *AE2* silencing. Our results indicated that the expression level of *NKCC1* gradually decreased after *AE2* was silenced. This finding suggests a regulatory relationship between *AE2* and *NKCC1*, which together play an important regulatory role in the nitrite stress process. Similar results have been observed in human airway epithelial cells. AE2 can regulate intracellular pH by transporting HCO_3_^−^ outside the membrane and synergizes with NKCC1 to promote intracellular Cl^−^ accumulation [32].

The activity levels of ACP and AKP can directly reflect the growth status of the organism, the immune function, and the ability to cope with changes in the external environment [33]. Our results demonstrated that ACP activity in the control group initially increased and then decreased. In contrast, the interference group exhibited a decline in activity. AKP activity and expression levels in the control group exhibited a gradual decrease, while enzyme activity in the interference group gradually increased. This suggests that both the control and interference groups were affected by the immune response, and that nitrite stress suppressed the immune functions of shrimp [34,35]. Interference with the *AE2* gene can reduce the impairment of shrimp immune function caused by nitrite to a certain extent. SOD and GST scavenge reactive oxygen species and prevent oxidative damage [36]. Numerous studies have shown that SOD and GST activities are significantly elevated after nitrite stress. However, with the increase in exposure concentration and time, the organism’s capacity for scavenging and repairing declines. The antioxidant enzyme activities diminish when substantial quantities of nitrite are accumulated in the body, leading to damage [37,38]. In the present study, SOD and GST activities and gene expression levels first increased and then decreased in the control group, which is consistent with the results of previous studies. In the interference group, the activity and gene expression level of GST gradually increased, while those of SOD firstly increased, and then decreased, gradually returning to normal levels. Therefore, after *AE2* was silenced, the amount of nitrite entering the shrimp’s body was reduced, and the organism’s scavenging and repairing abilities were not seriously affected. In this study, the interference group exhibited less damage to gill tissue compared to the control group, with no instances of cuticle fragmentation and secretion of eosinophilic substances observed. These findings suggest that silencing of *AE2* reduces nitrite damage to gill tissue.

## 4. Materials and Methods

### 4.1. Experimental Animals

A total of 500 healthy shrimp (average weight: 1.4 ± 0.3 g) were purchased from Shanghai Shencao Special Aquatic Products Development Co., Ltd. (Shanghai, China). The formal experiment was started after one week of temporary rearing in tanks (salinity 1.0 ± 0.1, pH 8.2 ± 0.2, water temperature 28 ± 2 °C).

### 4.2. RNA Extraction and cDNA Synthesis

Total RNA was extracted using the TRleasy Plus Total RNA Kit (Yeasen, Shanghai, China). The purity and integrity of the extracted RNA were then assessed using a Nano-300 (Aosheng, Hangzhou, China) and 1% agarose gel electrophoresis, respectively. The RNA was reverse-transcribed into cDNA using a reverse transcription kit (Yeasen).

### 4.3. Full-Length cDNA Cloning of the AE2 Gene from L. vannamei

The *AE2* gene sequence was obtained based on the transcriptome database of *L. vannamei* by our research group. Specific primers were designed based on the validated core sequences (Table 1). The first strand cDNA was synthesized using SMART 5′ RACE and 3′ RACE kits (Clontech, San Francisco, CA, USA). The 3′ and 5′ terminal cDNA sequences were amplified using the rapid amplification of cDNA ends (RACE) method, respectively. The PCR products were ligated into the pMD19-T vector and transformed into DH5α receptor cells. The positive monoclonal strains were sent to Shanghai Sangon Biotech Co., Ltd. (Shanghai, China), for sequencing.

### 4.4. Sequence Analysis of AE2 Gene

Based on the full length of cDNA obtained after splicing, bioinformatics analysis was performed according to the method in Table 2.

### 4.5. Tissue Expression Analysis of AE2 Gene

Heart, hepatopancreas, gill, eyestalk, muscle, intestine, stomach, testis, and ovary tissues were obtained separately for RNA extraction and cDNA synthesis. Tissue distribution of *AE2* was analyzed using RT-qPCR, with β-actin as an internal reference gene. Specific primers were designed based on the *AE2* gene sequence using Primer 5.0 software (Table 1). RT-qPCR amplification reactions were performed using the SYBR^®^ Green Premix Pro Taq HS qPCR Kit (Accurate Biology, Changsha, China). The assay was conducted in triplicate for each sample. The 2^−∆∆Ct^ method was employed to calculate the relative expression level of *AE2* [39].

### 4.6. In Situ Hybridization

The T7 promoter sequence (TAATACGACTCACTATAGGG) was incorporated at the 5′ end of the reverse primer to serve as an antisense probe, while it was added to the 5′ end of the forward primer as a sense probe (control group). The purified PCR products were transcribed in vitro using the T7 High Performance Transcription Kit (TransGen Biotech, Beijing, China), and labeled probes were obtained using DIG RNA Labeling Mix (Roche, Germany).

Gill tissue was placed in an in situ hybridization fixative (Sangon Biotech Co., Ltd.) and fixed for 24 h. Thereafter, fixed samples were subjected to dehydration via a graded alcohol series and transparency with xylene. Finally, samples were paraffin-embedded and sectioned to a thickness of ~5 μm. Next, in situ hybridization was conducted in accordance with the instructions provided with the DIG Nucleic Acid Detection Kit (SP6/T7; Roche), with BCIP/NBT chromogen treatment for light avoidance. The hybridization signal was observed and photographed using a Leica DM 2500 microscope (Leica, Heidelberg, Germany).

### 4.7. siRNA Interference

Based on the AE2 sequence, we designed and synthesized three pairs of siRNAs (G1, G2, and G3, respectively; Table 1). In this experiment, an equal amount of DEPC water was injected as a blank control group, NC chain was injected as a negative control group, and G1, G2, and G3 chains were injected as experimental groups. Three parallel groups included 15 shrimp per group. Injections were administered into the pericardial site via microsyringe, with an injection volume of 1 μg/g [40,41]. Gill tissue was obtained at 24, 48, 72, and 96 h, respectively.

### 4.8. Nitrite Stress Following AE2 Silencing by RNAi

Based on pre-experiment results, we selected the G2 and the 24 h time point after interference for the nitrite stress experiment. Different interference solutions were injected into the shrimp. After 24 h, the nitrite concentration in the water column was adjusted to the 96 h LC_50_ (16.50 mg/L, based on the results of previous experiments). The experimental design included six replicates with 10 shrimp per group, and three replicates used to calculate the survival rate at each time point. Gill tissue was obtained at 24, 48, 72, and 96 h post-exposure. One portion of each sample was fixed in 4% paraformaldehyde and processed for microscopic observation. The rest was rapidly frozen in liquid nitrogen and stored at −80 °C for subsequent analysis of gene expression and enzyme activity.

### 4.9. Determination of Nitrite Content in Hemolymph and Enzyme Activity in Gill Tissue

To determine the gill SOD, AKP, ACP, and GSH activities, 0.9% saline was added to the gill samples, which were then homogenized at low temperature to form a 10% tissue homogenate. This was centrifuged for 15 min at 4 °C and 12,000× *g*/min and the supernatant was collected for biochemical analysis using relevant kits purchased from Nanjing Jiancheng Co. following the manufacturer’s instructions.

The nitrite content in serum were determined using respective kits purchased from Nanjing Jiancheng Co. (Nanjing, China) following the manufacturer’s instructions. All indicators were detected using a Multiskan FC enzyme labeler (Thermo Fisher Scientific, Waltham, MA, USA) and 754N UV spectrophotometer (Yidian, Shanghai, China).

### 4.10. Determination of Expression Levels of Immune and Anti-Oxidative Stress-Related Genes

The expression levels of the *NKCC1*, *ACP*, *AKP*, and *SOD* genes were examined in each treatment group following exposure to nitrite stress. Specific primers were designed using Primer 5 (Table 1). RT-qPCR analysis consisted of three biological replicates and three technical replicates. The relative expression of each target gene relative to the β-actin gene was detected.

### 4.11. Microscopic Observation of Gill Tissue

Gill tissue was placed in 4% paraformaldehyde solution for 24 h for fixation, then subjected to ethanol gradient dehydration, transparent treatment, paraffin embedding, and sectioning. Sections were ~5 mm thick. Staining was performed using hematoxylin–eosin and neutral gum was used to seal the film. Tissue morphology was observed and photographed using an a DM1000 orthostatic microscope (Leica, Heidelberg, Germany).

### 4.12. Data Statistics and Analysis

Experimental data were analyzed using SPSS 18.0. Origin Pro 2021 was used to plot the images. One-way ANOVA was used to test for statistical differences. *p* < 0.05 was considered significant.

## 5. Conclusions

In this study, the full-length cDNA sequence of the *AE2* gene was identified and cloned from *L. vannamei*. We explored the gene’s sequence characteristics, tissue distribution, and the effects of nitrite on shrimp survival, physiology, and tissue damage following gene silencing. Interference with *AE2* can effectively improve the survival rate of shrimp, reduce the accumulation of nitrite in the body, and reduce the damage of nitrite to gill tissue. At the same time, the expression levels of *NKCC1* was reduced. These results of this study suggest that AE2 and NKCC1 are jointly involved in regulating the entry of nitrite into the shrimp body through gill tissue.

## Figures and Tables

**Figure 1 ijms-26-00964-f001:**
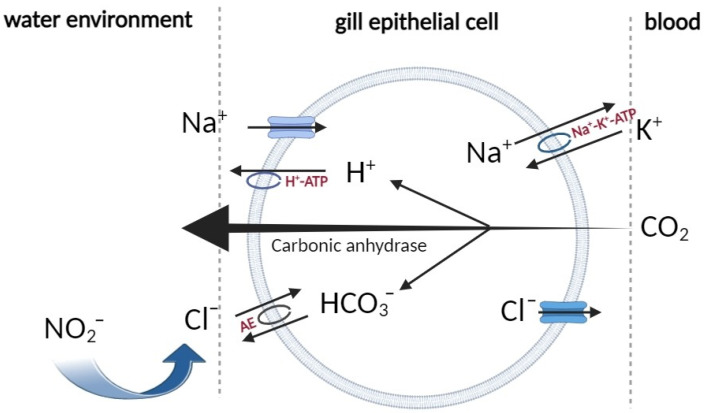
The mechanisms of nitrite uptake in freshwater fish.

**Figure 2 ijms-26-00964-f002:**
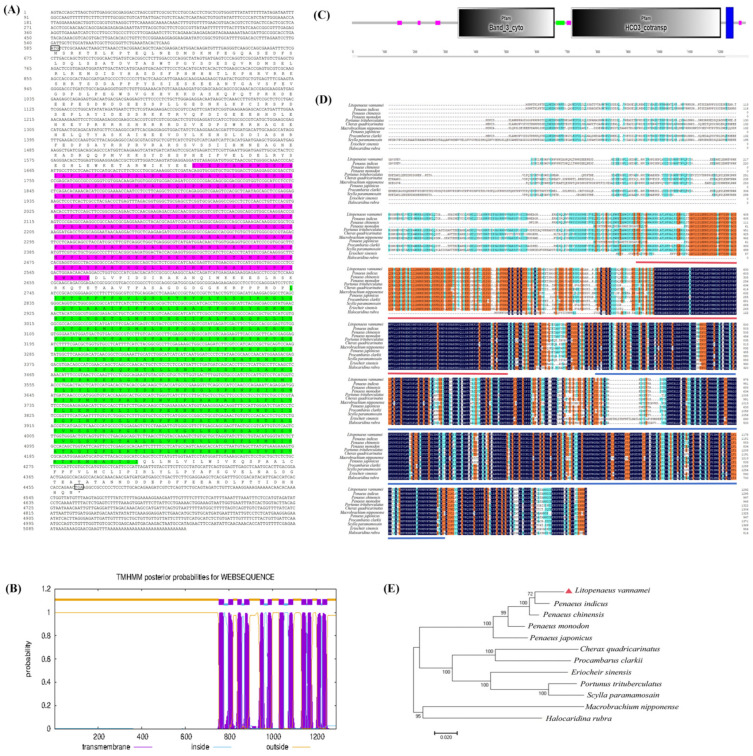
(**A**) Nucleotide and amino acid sequence of AE2. Black boxes indicate start and stop codons. Purple shading indicates Band3 structural domain. Green shading indicates HCO_3_^−^ structural domain. (**B**) Predicted transmembrane region of AE2. (**C**) Structural prediction of AE2. Purple regions indicate low complexity; green region indicates coiled coil region; blue region indicates transmembrane region. (**D**) Sequence comparison of homology of AE2 with other species with *L. vannamei*. Black regions indicate 100 per cent homology; orange regions indicate greater than 75 per cent homology; green regions indicate greater than 50 per cent homology. Other species NCBI accession numbers: Penaeus indicus: XP_063600464.1; Penaeus chinensis: XP_047502127.1; Penaeus monodon: XP_037800195.1; Portunus trituberculatus: XP_045109016.1; Cherax quadricarinatus: XP_053652588.1; Macrobrachium nipponense: XP_064098014.1; Penaeus japonicus: XP_042864807.1; Procambarus clarkii: XP_045584458.1; Scylla paramamosain: XP_063879958.1; Eriocheir sinensis: XP_050725191.1; Halocaridina rubra: KAK7074383.1. (**E**) Phylogenetic analysis of AE2 protein from different species.

**Figure 3 ijms-26-00964-f003:**
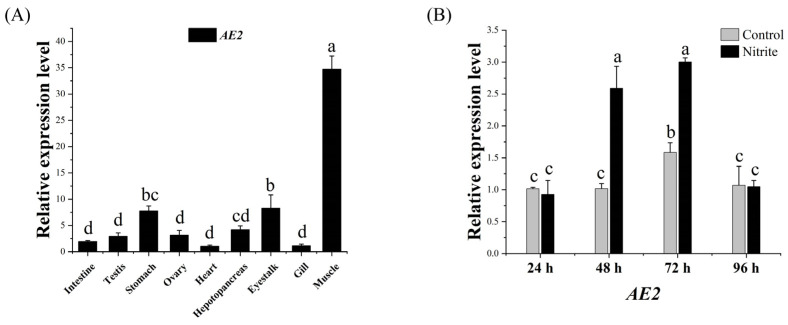
(**A**) Expression of the *AE2* gene in various tissues of *L. vannamei*. (**B**) Expression of *AE2* in gill tissue after nitrite stress. Different letters indicate significant differences (*p* < 0.05).

**Figure 4 ijms-26-00964-f004:**
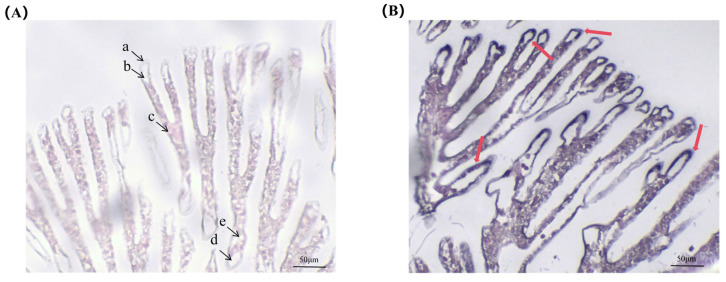
In situ hybridization analysis of gill tissue of *L. vannamei*. (**A**) Localization results of sense probe in gill tissue. (**B**) Localization results of antisense probe in gill tissue. a: cuticle; b: epithelial cells; c: hemolymph cells; d: entering gill vessels; e: exiting gill vessels. Red arrows indicate positive signals.

**Figure 5 ijms-26-00964-f005:**
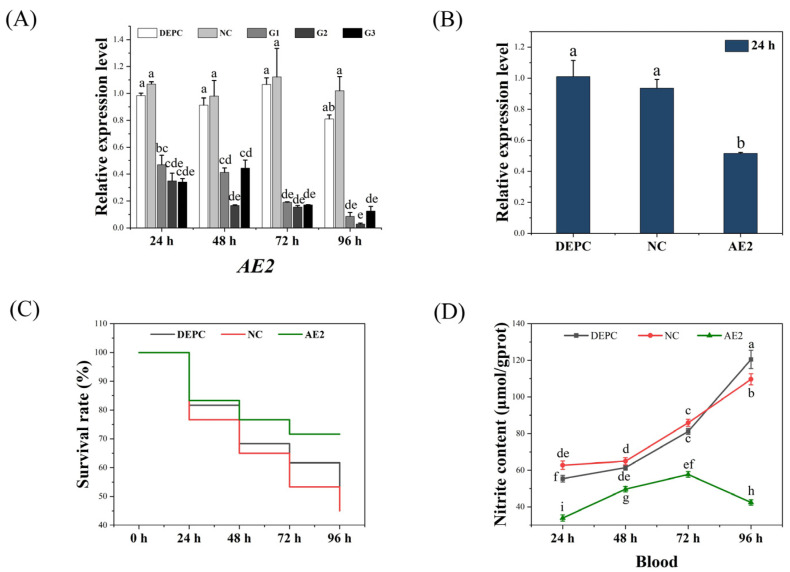
(**A**) Expression level of *AE2* in gill tissue after RNAi. (**B**) Expression level of *AE2* after injection of G2. Effects of nitrite stress on survival rate (**C**) and hemolymph nitrite level (**D**) of shrimp after RNAi. DEPC: diethyl pyrocarbonate; NC: negative control, nonsense sequence from nematode. Different letters indicate significant differences (*p* < 0.05).

**Figure 6 ijms-26-00964-f006:**
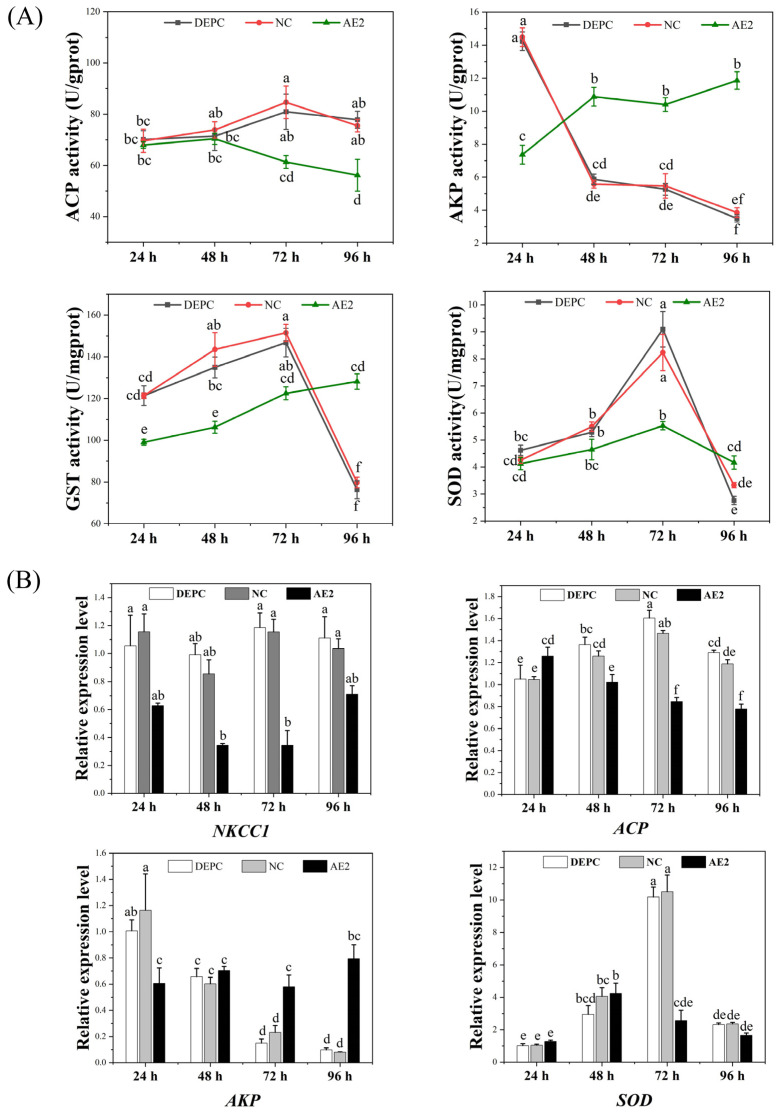
Changes in immune and anti-oxidative stress-related enzyme activities (**A**) and gene expression (**B**). Different letters indicate significant differences (*p* < 0.05).

**Figure 7 ijms-26-00964-f007:**
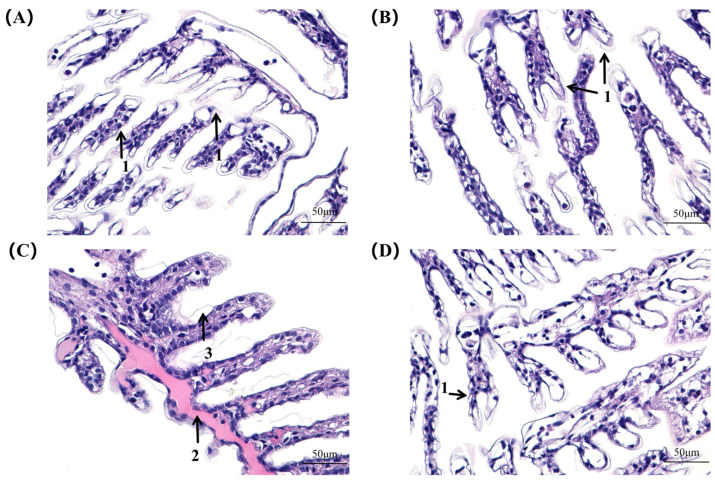
Microscopic observation of gill tissue in the control (**A**) and experimental (**B**) groups 24 h after nitrite stress. Microscopic observation of gill tissue in control (**C**) and experimental (**D**) groups 96 h after nitrite stress. Separation of stratum cuticle and epithelial cells—1; secretion of eosinophilic substances—2; fragmentation of stratum cuticle structure—3.

**Table 1 ijms-26-00964-t001:** Primer sequences used in this experiment.

Primer	Sequence (5′-3′)	Purpose
AE2-1	F: GGGTCAAGCCAGCGAAGA	Sequence verification
	R: CCCAGCGGTTAGCCACAT	
AE2-2	F: GGAGGATGTAGAGGATGTGGC	
	R: CGGACAGGTAGAGCGGGTAG	
AE2-3	F: GCTACCCGCTCTACCTGTCC	
	R: GTCAATGTATGTCGGCAAATC	
AE2-5-outer	TCTGAGCTGAGACTCCGTGA	5′RACE
AE2-5-inner	ACATCAGTTGCCGAGGACAG	
AE2-3-outer	GACAAACAGCCATGATTCAGTGT	3′RACE
AE2-3-inner	GGGAGATTGATTGTTTTGCTGC	
AE2-RT	F: CGGCTCGGAAATTGCATCTG	RT-qPCR
	R: GCTTGTCGGTGTAGGTGTCA	
AE2- ISH-sense	F: TAATACGACTCACTATAGGGCGGCTCGGAAATTGCATCTG	In situ hybridization
	R: GCTTGTCGGTGTAGGTGTCA	
AE2- ISH-antisense	F: CGGCTCGGAAATTGCATCTG	
	R: TAATACGACTCACTATAGGGGCTTGTCGGTGTAGGTGTCA	
SiRNA-G1	F: AUGAAUCUUCUCGUAAGAATT	RNA interference
	R: UUCUUACGAGAAGAUUCAUTT	
SiRNA-G2	F: AAGAAGUCAUAUGAUCAUATT	
	R: UAUGAUCAUAUGACUUCUUTT	
SiRNA-G3	F: GCGAGUUAGUGCCGUCAUUTT	
	R: AAUGACGGCACUAACUCGCTT	
SiRNA-negative control	F: UUCUCCGAACGUGUCACGUTT	
	R: ACGUGACACGUUCGGAGAATT	
Acid phosphatase XM_027370834.1	F: CTCGGATAATGCTCGTGTCG	
	R: TGCTGAATCTTGCTCTGTAGTTG	
Alkaline phosphatase XM_027360250.1	F: GAACCGCAATGCTGTAGAAG	
	R: CGCTGTAGGTCTTGATGAGTG	
Superoxide dismutase XM_027383584.1	F: GACACGACCATTAGCCTGTACGAC	
	R: GTTGCCAGTAGCGAGTGAACCTTC	
Na^+^/K^+^/2Cl^−^ PQ073211	TGGACGGAGGTCTCAATG	
	CCAGAAGTCAAGCCTACAA	
β-actin XM_027364954.1	F: GCCCTGTTCCAGCCCTCATT	
	R: ACGGATGTCCACGTCGCACT	

**Table 2 ijms-26-00964-t002:** Bioinformatics analysis of AE2.

Method of Prediction	Applications
BLAST (https://www.ncbi.nlm.gov/blast (accessed on 15 July 2024))	Comparing amino acid sequence homology
ORF finder (http://www.ncbi.nlm.nih.gov/orffinder/ (accessed on 15 July 2024))	Predicting the open reading frame (ORF) position of the gene
Expasy-ProtParam (http://web.expasy.org/protparam/ (accessed on 15 July 2024))	Predicting the relative molecular weight and isoelectric point of the protein
SignalP 3.0 Server (http://www.cbs.dtu.dk/services/SignalP/ (accessed on 15 July 2024))	Predicting signal peptides
SMART (http://smart.embl-heidelberg.de/ (accessed on 15 July 2024))	Predicting and identifying functional structural domains.
TMHMM SerVer.2.0 (http://www.cbs.dtu.dk/services/TMHMM/ (accessed on 15 July 2024))	Predicting transmembrane structures
DNAMAN	Perform multiple comparisons of amino acid sequences
MEGA 7.0	Constructing a phylogenetic tree

## Data Availability

All the data presented in this study are included in the article.

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
