# Peer review of "Molecular Characterization of Anion Exchanger 2 in Litopenaeus vannamei and Its Role in Nitrite Stress"

_ijms, 2025, doi:10.3390/ijms26030964_

Round 1
Reviewer 1 Report
Comments and Suggestions for Authors
Dear Authors,
I believe the paper is interesting, well-written, and the data is clear, so it deserves to be published. I only have a few minor comments to make.
-Indicate in the caption of Figure 5 what DEPC and NC mean, please.
-I don't understand why in Fig 5D the nitrite content at the 96-hour point is lower than at 48 and 72 hours. Please provide a reasoned explanation.
-In Fig. 5, why was the expression level of GST not studied? Please provide an explanation.
What is the homology between AE1, AE2, and AE3? The regions of AE2 selected for srRNA G1, G2, and G3 are specific concerning AE1 and AE3?
L291: 4.4. Sequence analysis of4.4. Sequence analysis of AE2 gene. Typo
L348: “After 24h, the nitrite concentration in the water column was adjusted to the 96 h LC50 (the 96 h LC50 of 16.50 mg/L was obtained based on the results of previous experiments)” It's not clear to me, on what basis did you decide to use a specific concentration of nitrite? What was the initial concentration?
-Could nitrite have any beneficial effects on animals? Is there anything specific known about shrimp? Could nitrite be converted into nitric oxide? Could the toxic effect of nitrite be due to an excess of nitric oxide? Please discus
Reviewer 2 Report
Comments and Suggestions for Authors
A very similar, almost identical, manuscript (MS) was submitted recently to Comp. Biochem. Pysiol. As a reviewer of both manuscripts, I can detect the differences and similarities between them. While the first MS focused to Na/K/2Cl (NKCC) co-transporter and the competition between Cl and NO2 transport mainly across the gills (MS1), the second one analyzes the role of an anion exchanger (AE2) involving Cl and HCO3 exchange for which a competition between NO2 is also hypothesized and analyzed (MS2). The main problem is that most portions of both MS are identical. First, of all this should not be allowed and can be easily detected by classical software analyzing duplicity. Second, the main question is probably different and therefore cannot be addressed in the same way. Specific comments to these points are developed below for each section, but the MS cannot be accepted unless a clear distinction between both MS is done.
Abstract
This section is identical for MS1 and MS2, only NKCC was substituted by AE2
Introduction
Two out of three paragraphs are identical for MS1 and MS2, only NKCC was substituted by AE2
While a useful diagram is included in the second paragraph describing specifically the role of AE2 exchanger, no relation with the NKCC co-transporter explored in MS1 is considered. This should be done as both mechanisms are involved in NO2 entry via competition with Cl active uptake.
Line 58-59 revise term basal parietal membrane. I think it should be apical membrane
Material and methods
Same comment: the whole section is identical for MS and MS2, only NKCC was substituted by AE2 and G1 was substituted by G2 chain for RNA interference.
Comments addressed to MS1 also applies here:
Table 1, line 335-336, line 346-347: I am sorry, but I do not understand what the NC chain is (no explanation of abbreviation is provided) and in which way it represents a negative control. Similarly other explanations of abbreviations are needed: DEPC (line 335 and 346) and all abbreviations in Table 1 even if these are defined in text. Explain why two control groups were included.
Results
Of course, the results per se are different between MS1 and MS2 (except for Fig 5, see below), but the organization and data presentation are still identical.
It should be noted that the higher expression of AE2 was obtained in muscle, with minimal expression in gill; whereas for NKCC, gill exhibit the highest expression and justify why in situ hybridization was analyzed in this tissue. Such differences are important for the discussion section.
An important concern is that data in Fig 5 regarding both control groups (DEPC and NC) are the same than in MS1.
Comments addressed to MS1 also applies here:
Figure 5 should be split in two separate figures (interference and antioxidant enzymes) as in the present format they are too small.
Discussion
This section shows a lower degree of identity between MS1 and MS2, because different mechanisms are analyzed, although some sentences (and even entire paragraphs) are still identical.
Line 200-201. Check if AE2 is really located in basolateral membrane of epithelial cells as from Fig. 1 it seems that is the apical membrane.
Line 204-207 The explanation of a higher expression of AE2 in muscle related is scarce (“to support physiological process”) and must be better explained, probably due a higher metabolic activity that implies and active elimination of HCO3. In any case, it does not seem to be related to NO2 toxicity, although this was not assessed (i.e. AE2 expression in muscle following NO2 exposure). Why? In contrast, and although the expression in gills was one of the lowest among the tissues analyzed, this is the tissue in which NO2 competes with Cl and explains NO2 toxicity as shown by several results (AE2 expression in relation to NO2 exposure, AE2 interference). However, this is not explained: while the gill AE2 expression is lower than in other tissues, its role in NO2 toxicity is predominant.
Line 208-209 The statement do not seem to correspond with the cited reference. I was interested to check this reference as at high salinities the competition between Cl and NO2 uptake should not exist; on the contrary, at such salinities Cl should be eliminated.
Line 217-228 This paragraph is again almost identical to MS1 focused to NKCC. Instead of just repeating the same ideas that evidences the role of NKCC (MS1) and AE2 (MS2) in NO2 toxicity, authors should compare both mechanisms are they are closely related in terms of Cl uptake at low salinities and competition between Cl and NO2.
Same comments for the rest of the discussion: the two paragraphs from lines 229-255 and 256-263 about the implications of NO2 toxicity are identical between MS1 and MS2, without any discussion and comparisons between both mechanisms involved in NO2 uptake (AE2 and NKCC).
Round 2
Reviewer 2 Report
Comments and Suggestions for Authors
Authors have adequately revised their manuscript and made a big effort to distinguish both manuscript and present now a complementary approach of the role of both mechanisms (NKCC and AE2) involved. Just a minor correction I have detected:
Line 204-205 does not PRESENT a high ….., WHEREAS elevated expression
